# Comparison between Conventional Modality Versus Cone-Beam Computer Tomography on the Assessment of Vertical Furcation in Molars

**DOI:** 10.3390/diagnostics13010106

**Published:** 2022-12-29

**Authors:** Jack Lam, Andy Wai Kan Yeung, Aneesha Acharya, Chris Fok, Melissa Fok, Georgios Pelekos

**Affiliations:** 1Division of Periodontology and Implant Dentistry, Faculty of Dentistry, The University of Hong Kong, Hong Kong SAR, China; 2Division of Applied Oral Sciences & Community Dental Care, The University of Hong Kong, Hong Kong SAR, China; 3Dr. D.Y. Patil Dental College and Hospital, Pune 411018, Maharashtra, India

**Keywords:** furcation, vertical subclassification, cone-beam computer tomography, periodontal diagnostics

## Abstract

This study aimed to assess the accuracy of diagnosis of vertical furcation subclass in molars using periapical radiographs (PAs) and clinical chartings compared against cone-beam computer tomography (CBCT) as the gold standard. The protocol involved examiners with different levels of experience. This retrospective radiographic study retrieved 40 molar teeth with full periodontal chartings, PAs, and CBCT records. Fifteen examiners with different levels of experience evaluated the PAs and periodontal chartings to assess the vertical depth of furcation and, thus, the vertical subclassification. CBCT was used as the gold standard for comparison. The accuracy of vertical furcal depth measured was assessed together with the accuracy of vertical subclassification assignment. The reliability of the conventional diagnostic modality among the examiners was also evaluated. A linear mixed model adjusted for the CBCT vertical furcal depth measurement was constructed to determine if tooth position, horizontal furcation distribution, and examiner experience level affect the bias in the vertical depth of furcation measurement. The reliability of the conventional periodontal diagnostic method in measuring vertical furcal depth was found to be fair, while vertical subclass assignment was moderate. Significantly better reliability during subclass assignment was found with mandibular molars (*p* < 0.001) and in maxillary molars with isolated buccal class II furcation. Within the study’s limitations, conventional periodontal diagnostics based on periapical radiographs and clinical periodontal chartings appear to be in poor to fair agreement with CBCT (gold standard) when measuring the vertical depth of furcation. Examiners with the least experience were more prone to bias when estimating the vertical furcal depth.

## 1. Introduction

Advanced periodontal disease may be presented with destruction extending into the furcation area of multirooted teeth, leading to furcation involvement (FI). The classic Hamp classification differentiates FI into different degrees depending on the amount of horizontal loss of periodontal tissue support measured with a graduated Nabers furcation probe [1]. According to previous retrospective studies and systematic reviews, most molars with FI respond well to periodontal therapy and could be maintained successfully if supportive periodontal care is provided [2,3]. In the vertical dimension, FI was classified initially into subclasses A, B, and C, with a vertical loss of attachment of 0–3 mm, 4–6 mm, and more than or equal to 7 mm, respectively [4]. A modification was proposed recently with subclasses A, B, and C redefined as attachment loss extending to the coronal, middle, and apical third of the roots, respectively [5].

The degree of vertical furcation subclassification has been shown to affect molars’ long-term prognosis in a retrospective cohort study involving 200 periodontal patients who underwent supportive periodontal care for at least ten years. Ten-year survival rates of 91%, 67%, and 23% were reported for molars with vertical subclasses A, B, and C [5]. In another retrospective study, molars with FI were around five times more likely to be extracted than molars with no FI in the multivariable analysis. This risk was higher when the vertical furcation component was considered [6]. Vertical classification allows for the assessment of the indications for planning different furcation treatment options, such as open flap debridement and resective or regenerative periodontal surgeries. In cases with advanced furcal destruction, periodontal regeneration could increase periodontal support and improve tooth prognosis by reducing the furcation subclassification [7]. Vertical subclassification is vital in determining molars’ prognosis, deciding on appropriate treatment, and customising supportive care to aid their long-term retention.

A retrospective study compared baseline probing, bone sounding, and intra-surgical vertical and horizontal measurements of furcation defects in molars that received open flap surgeries [8]. Pre-anesthesia probing was the same in 42% and 47.1% of sites concerning the intra-surgical assessment in the vertical and horizontal dimensions. Bone sounding improved the agreement to 59.5% and 64.2%. 

Plain radiographs were shown to be in moderate agreement with intra-surgical measurement in terms of horizontal furcation classification [9]. The use of “furcation arrows” on periapical radiographs as an indicator of the presence of FI was found to be of low sensitivity (38.4%) [10]. However, the reliability and validity of PA radiographs in the vertical assessment of furcation have yet to be determined. 

A previous in vitro study has demonstrated high consistency between computer tomography and histological sections in classifying horizontal and vertical FI, with jaw specimens’ defects artificially created [11]. Another prospective clinical study involving 20 maxillary molars has demonstrated strong agreement between CBCT and intra-surgical measurements of horizontal furcation degree and vertical bone loss, with a difference of 0.36 mm [12]. Therefore, a high correspondence between CBCT and intra-surgical measurement, which is classically deemed as the gold standard, could be expected [13]. Furthermore, CBCT images have been shown to improve the treatment decision-making process and optimise treatment planning in maxillary molars [14,15]. A recent systematic review that explored the evaluation of FI by different diagnostic methods reported that CBCT displayed the closest agreement with intra-surgical measurements, while the accuracy of all the methods was influenced by the examiner’s experience [16].

Despite its high consistency and reliability, CBCT is not routinely performed to assess furcation due to concern with the extra radiation dose imposed, compared to traditional plain radiographs. On the other hand, the acquisition of CBCT images using a low-dose protocol, a shorter exposure time, and a smaller field of view could be performed to minimise the effective radiation dose of CBCT [17]. Therefore, this study aimed to assess if diagnosing vertical furcation subclass in molars using conventional periodontal examination methods is reliable and valid. Additionally, the effects of tooth position, furcation involvement distribution, and examiner experience level were investigated.

## 2. Materials and Methods

### 2.1. Study Design and Database Search

The current study is a preliminary, retrospective radiographic study comparing the conventional periodontal diagnostic method and cone-beam computer tomography to assess molar furcation involvement in the vertical dimension. A database search of periodontal patients who received care in the periodontology division was performed by the same single calibrated investigator that performed the clinical measurements (GP) to obtain records of molar teeth that met the inclusion criteria. Forty teeth were selected using a convenience sampling method. The study was approved by the Institutional Review Board of the University of Hong Kong/Hospital Authority Hong Kong West Cluster.

### 2.2. Inclusion and Exclusion Criteria

The inclusion criteria were (i) maxillary or mandibular molars with at least class II furcation involvement and (ii) 6-point clinical periodontal chartings, periapical radiographs, and CBCT DICOM files. Exclusion criteria in this study include (i) poor X-ray and CBCT quality; (ii) periodontal bone loss beyond the apex; (iii) endodontic/periapical pathology; (iv) cracked tooth syndrome/root fracture; (v) molars with fused roots; (vi) presence of fremitus or secondary occlusal trauma; (vii) presence of distolingual root in mandibular molars; (viii) presence of extra mesiobuccal root in maxillary molars; and (ix) presence of extensive metallic restorations close to the furcation area. 

### 2.3. Radiographic Records

All digital periapical radiographs were taken with Durr digital phosphor plates and exposed with an intraoral X-ray machine, Veraview IX (J. Morita, Tokyo, Japan; exposure settings: 60 kV, 7 mA; exposure times: 0.1–0,2 s). Images were imported into the Durr VistaScan Mini view machine (Durr Dental, Stuttgart, Germany). CBCT scanning was performed with either a Veraview X800 unit (J. Morita, Tokyo, Japan; field of view (FOV, diameter x height): 100 × 80 mm; exposure settings: 100 kV, 5 mA; voxel size: 0.125 mm) or ProMax 3D Classic (Planmeca, Helsinki, Finland; FOV: 100 × 100 mm; exposure settings: 90 kV, 8 mA; voxel size: 0.200 mm). All the radiographic records were evaluated by an experienced dental radiologist (AWKY) from the Department of Oral and Maxillofacial Radiology, Faculty of Dentistry, HKU. Radiographs and CBCT images were evaluated for film bending, tilting/angulation error, contrast and brightness, movement during capture, metallic scattering, and artefacts.

### 2.4. Gold Standard Assessment

CBCT DICOM files of the molars included were imported into the Romexis Viewer (Version 6.1, Planmeca, Helsinki, Finland) and analysed by two calibrated investigators (P.M.L. and A.W.K.Y.). The horizontal furcation involvement, as shown in the clinical charting, was first confirmed on the CBCT images. An assessment of the vertical bone loss in the furcation was then performed. The CBCT viewer axis was first aligned along the long axis of the tooth at the sagittal section and coronal section. At the axial section, the sagittal and coronal planes were adjusted to intersect the canal orifices of the two roots bounding the concerned furcation involvement (Figure 1). The sites with a deeper vertical furcal depth were chosen. The measurement was made to the nearest 0.1 mm using the linear measurement tool on the Romexis Viewer. The tooth was classified according to the modified vertical furcation subclassification and served as the gold standard. By toggling the CBCT along the adjusted axis, the maximal vertical distance from the furcation fornix to the base of the defect and the length of the shorter root bounding the furcation were measured. The exact measurement was repeated on the other furcation entrance(s) if the subject tooth had more than one class II horizontal FI in the maxillary molars (Figure 2). 

### 2.5. Examiners and Image Analysis

In the last decade, CBCT was introduced in the undergraduate curriculum of this institution (HKU) as a new diagnostic modality with increasing popularity due to its three-dimensional nature. During postgraduate training, further practical and theoretical training takes place so that CBCT usage as a diagnostic and treatment planning tool is enhanced. Fifteen examiners consisting of five specialist periodontists (level 3 experience), five postgraduate trainees in the Department of Periodontology and Implant Dentistry (level 2 experience), and five junior hospital dental officers (level 1 experience) were recruited. 

The examiners were briefed on the study procedures and the furcation classification systems used in the assessment. Each examiner was provided with clinical periodontal chartings and periapical radiographs. All periapical radiographs were imported into the Romexis Viewer software (Planmeca) and were analysed on the same computer screen (Microsoft Surface Pro 7). On the periapical radiographs, the examiners were asked to measure the vertical bone loss in the furcation area (to the nearest 0.1 mm) using the linear measurement tool in Romexis Viewer with the clinical periodontal charting as a reference (Figure 1 and Figure 3A). To diagnose the FI according to the modified vertical subclassification, the examiners were also asked to measure the length of the shorter root bounding the furcation (which was pre-determined as part of the gold standard assessment) (Figure 3B). All data were recorded on a data collection form in Microsoft Excel(Excel 2016 Inc, Redmond, Washington, USA).

### 2.6. Statistical Analysis

The data collection form in Microsoft Excel was proofed for entry error and inputted into the statistical program SPSS 28.0 (IBM, Armonk, NY, USA) for statistical analysis. The significance of differences in the depth of vertical furcal bone loss between the conventional modality measurement and the gold standard of each examiner was computed with ICC for (i) all 40 teeth, (ii) maxillary molars, and (iii) mandibular molars. Similarly, the significance of differences in the vertical furcation subclassification determined using the conventional modality against the gold standard of each examiner was computed with LWK. The reliability of the conventional modality in determining the vertical depth of furcation involvement and vertical subclassification among the 15 examiners was calculated using the ICC and Fleiss’ multirater kappa. All 40 teeth were included and subdivided according to the tooth position (maxillary/mandibular) and the horizontal furcation involvement distribution. 

The ICC and Fleiss’ kappa between the subgroups were compared using the *F*-test and *Z*-test, respectively, at a significance level of 0.05. 

To explore if tooth position (maxillary/mandibular), FI distribution, and examiner experience level affect the bias in measuring the vertical depth of FI, a linear mixed model considering both random intercept and repeated measurements by raters on 40 teeth was performed.

## 3. Results

Among the 40 selected subject teeth, 20 were maxillary molars, and 20 were mandibular molars. Four maxillary molars had isolated class II horizontal involvement at the buccal furcation. Eight had isolated class II horizontal involvement at either the mesial or distal furcation, and eight had at least two class II or class III horizontal furcation involvements. On the other hand, ten mandibular molars were presented with class II horizontal involvement at either the buccal or lingual furcation, and ten had class II involvement on both sides or class III furcation involvement. When compared against the gold standard vertical depth as measured on CBCT, the 15 examiners displayed an intraclass correlation coefficient range of 0.133–0.570. A higher ICC value could be observed in mandibular molars analysed individually, although statistical significance was only reached in the *F*-test analysis from 5 of the 15 examiners (Table 1). When vertical subclassification assignment was compared against CBCT, the examiners displayed an LWK range of 0.071–0.350. Similarly, when maxillary and mandibular molars were analysed individually, a general trend of a higher LWK value could be observed in mandibular molars. However, statistical significance was only reached in the *Z*-test analysis from 5 out of the 15 examiners (Table 2).

The reliability in measuring the vertical depth of FI conventionally among the 15 examiners was also assessed with an ICC value of 0.560 (95% CI 0.446–0.686), thus suggesting a fair agreement between the examiners [18]. A similar agreement was also found when maxillary and mandibular molars were considered separately, with an ICC value of 0.497 (95% CI 0.343–0.689) and 0.573 (95% CI 0.415–0.750), respectively. No statistical significance was found when the two ICC values were compared (*p* = 0.362). Interestingly when maxillary molars were assessed according to their FI topography, ICC values varied between 0.491 (95% CI 0.199–0.934) for isolated buccal class II FI, 0.470 (95% CI 0.250–0.796) for isolated mesial or distal class II FI, and 0.624 (95% CI 0.395–0.877) in the case of two class II or class III FIs. As for the mandibular molars, ICC values of 0.544 (95% CI 0.334–0.807) in the case of isolated buccal/lingual class II FI and 0.505 (95% CI 0.297–0.783) for buccal and lingual class II or class III FI were found. No statistically significant difference was found under *F*-test analysis for all subgroups in both the upper and lower molars (Table 3).

The reliability of the conventional assessment of subclass vertical subclassification among the 15 examiners was assessed with a Fleiss kappa value of 0.452 (95% CI 0.428–0.477), suggesting a moderate agreement between the examiners [19]. When maxillary and mandibular molars were considered separately, Fleiss kappa values of 0.204 (95% CI 0.172–0.237) and 0.568 (95% CI 0.529–0.608) were found, suggesting a fair agreement when diagnosing maxillary molars and a moderate agreement when diagnosing mandibular molars. The better agreement when diagnosing mandibular molars was statistically significant when the two Fleiss kappa values were compared using the *Z*-test (*p* < 0.001). 

Statistically significantly better agreement among the examiners was found in the maxillary isolated buccal class II furcation involvement subgroup compared to the other two subgroups using the *Z*-test (*p* = 0.004 and <0.001). 

As for the mandibular molars, statistically significantly better agreement among the examiners was found in the subgroup with both buccal and lingual class II or class III FI when compared using the *Z*-test (*p* = 0.014).

A linear mixed model (LMM) was constructed considering the estimated mean difference in the vertical depth of furcation involvement between the conventional modality and CBCT. Cofactors such as the tooth position (maxillary/mandibular), furcation involvement distribution, and examiner experience level were included in the model (Table 4).

Statistically significantly higher bias was observed in maxillary molars exhibiting isolated buccal class II FI compared to maxillary molars with isolated mesial/distal class II FI using Bonferroni multiple comparisons (*p* = 0.016).

The examiner experience level also appeared to affect the bias in measuring the vertical depth of furcation involvement (*p* < 0.01). It was observed that examiners who were recent graduates with less than one year of postgraduate experience (Level 1) had a significantly higher bias when compared to postgraduate clinicians with at least two years of postgraduate experience (Level 2) (*p* < 0.001) and specialist periodontists (Level 3) (*p* = 0.004), as revealed in the Bonferroni multiple comparisons. No statistically significant difference was found among the Level 2 and Level 3 examiners (*p* = 0.166).

Lastly, a statistically significant difference was detected when descriptive statistics and one-way ANOVA were used to compare the vertical depth of FI in different furcation involvement distribution subgroups as measured on CBCT (*p* < 0.001) (Table 5). The LMM was therefore adjusted and re-analysed, but in the adjusted model (Table 6), no statistically significant difference in bias was observed in maxillary molars and mandibular molars presented with different furcation involvement distributions (*p* = 0.228). The examiner experience level still appeared to affect the bias in measuring the vertical depth of furcation involvement (*p* < 0.001). Bonferroni multiple comparisons revealed a significantly higher bias in examiners with Level 1 experience compared to the more experienced Level 2 and 3 examiners (*p* < 0.001 and *p* = 0.03, respectively). No significant difference was observed between Level 2 and 3 examiners (*p* = 0.182).

## 4. Discussion

There are several risks using conventional clinical and radiographic examination, which could prevent the accurate diagnosis of FI, including gingival tissue consistency, inflammation severity, pressure while probing, probe size, probing angulation, and presence of dental restorations. A recent systematic review showed that CBCT was found to be advantageous and accurate in cases of infra-bony defects and FI [20].

In this preliminary study, vertical furcation assessment of 40 molar teeth was performed by 15 examiners using the conventional periodontal diagnostic methods. The agreement in measuring the vertical depth of FI and subclass assignment against CBCT by the 15 examiners was found to be poor to fair. When maxillary and mandibular molars were analysed separately, a general trend of higher ICC and linear weighted kappa values could be observed in mandibular molars. This is partly in agreement with a previous study in which the authors reported a significant difference in the depth of vertical furcal bone loss measured on pre-operative radiographs and intra-surgically in maxillary molars, but not in mandibular molars [21]. 

During assessment by CBCT, slides showing the maximal vertical furcal bone loss may not constantly occur at the furcation entrance. Intra-surgical measurement theoretically only allows vertical assessment to be conducted at the entrance. It may also be affected by factors such as probe angulation, the position of the furcation entrance, and intra-surgical visibility. Periodontal probes could only provide measurements to the nearest 0.5 mm. This contrasts the measurement tool used in CBCT diagnostic software, in which measurements could be made to the nearest 0.01 mm. This was shown clearly in a retrospective study that concluded that CBCT is validated as a valuable supplemental tool for assessment of molar FI in addition to periodontal probing and intraoral radiographic examinations [22]. Whether CBCT or intra-surgical measurements could better represent the actual vertical furcal bone loss remains a topic to be explored.

Additionally, significantly better agreement in vertical subclass was found among maxillary molars with isolated buccal class II furcation and mandibular molars with both buccal and lingual class II furcation involvement or class III furcation involvement.

A linear mixed model was constructed to explore the relationship between different factors. The model was subsequently adjusted, including the tooth position (maxillary/mandibular), furcation involvement distribution, and examiner experience level. No significant difference in bias was observed between molars of different furcation involvement patterns. Higher bias was observed in examiners with less than one year of postgraduate experience compared to the more experienced clinicians in both the unadjusted and adjusted models. This agrees with a previous study in which a significantly higher horizontal furcation degree assessment accuracy using intraoral radiographs was demonstrated by a more experienced examiner [9].

One major limitation of the current study is the limited sample size. Since 40 molars were included, specific subgroups (maxillary molars with isolated class II buccal furcation involvement) had a much smaller representation when compared to the other subgroups.

Due to the study’s retrospective nature, only 6-point periodontal chartings of the subject teeth could be retrieved from the database record; this may not be sufficient to reflect the actual attachment levels around a molar tooth. A difference in attachment levels may exist between the two roots bounding the same furcation, so a clinical assessment of individual roots may provide a better representation.

The current study found poor to fair agreement between the conventional diagnostic modality and CBCT in both the vertical furcal depth and subclassification assignment. Future research may also explore the reliability of using CBCT in the vertical assessment of furcation, and thus a comparison could be performed. Performing a study similar to the current study but with a prospective design and a larger sample size may also increase the power during the analysis.

## 5. Conclusions

Within the study’s limitations, conventional periodontal diagnostic methods based on periapical radiographs and clinical periodontal parameters appear to be in poor to fair agreement with CBCT in measuring the vertical depth of furcation and diagnosing molars according to the modified vertical subclassification. With conventional periodontal diagnostics, fair reliability was found for vertical furcal depth measurement, and moderate reliability was found for vertical subclass assignment. Examiners with the least experience exhibited greater bias when measuring the vertical furcal depth. In summary, it can be suggested that CBCT is beneficial in cases of molars with FI, especially when it comes to surgical treatment planning.

## Figures and Tables

**Figure 1 diagnostics-13-00106-f001:**
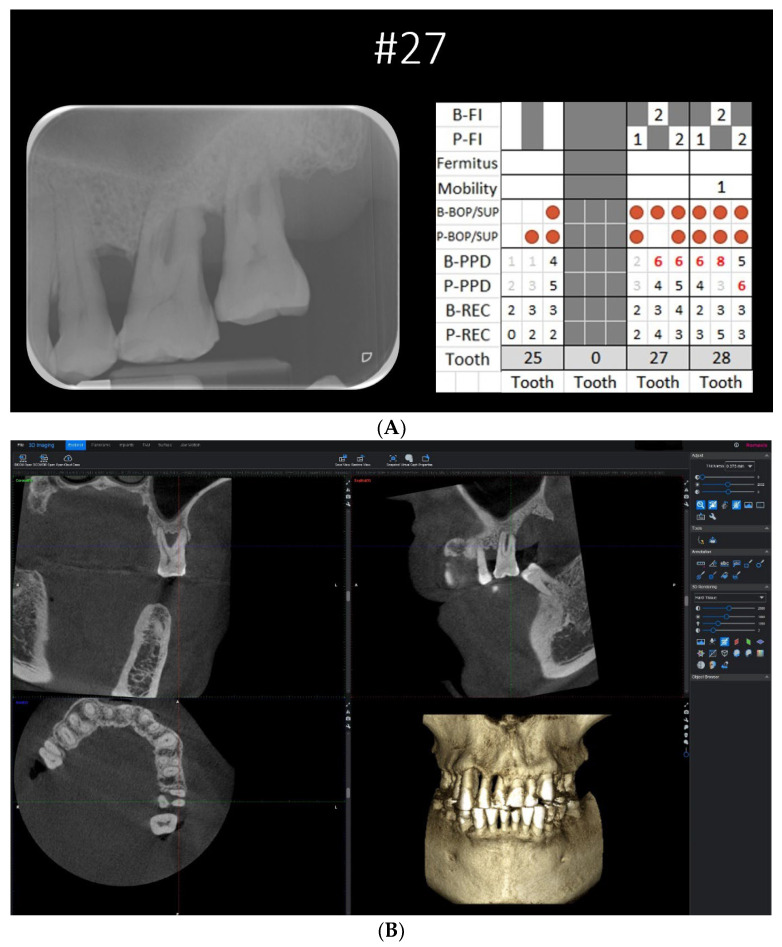
(**A**) Case example with a complete set of clinical periodontal record, periapical radiograph, and CBCT DICOM files as viewed on Romexis Viewer (Planmeca). (**B**) The CBCT viewer axis was first aligned along the long axis of the tooth at the sagittal section and coronal section. At the axial section, the sagittal and coronal planes were adjusted to intersect the canal orifices of the two roots bounding the concerned furcation involvement.

**Figure 2 diagnostics-13-00106-f002:**
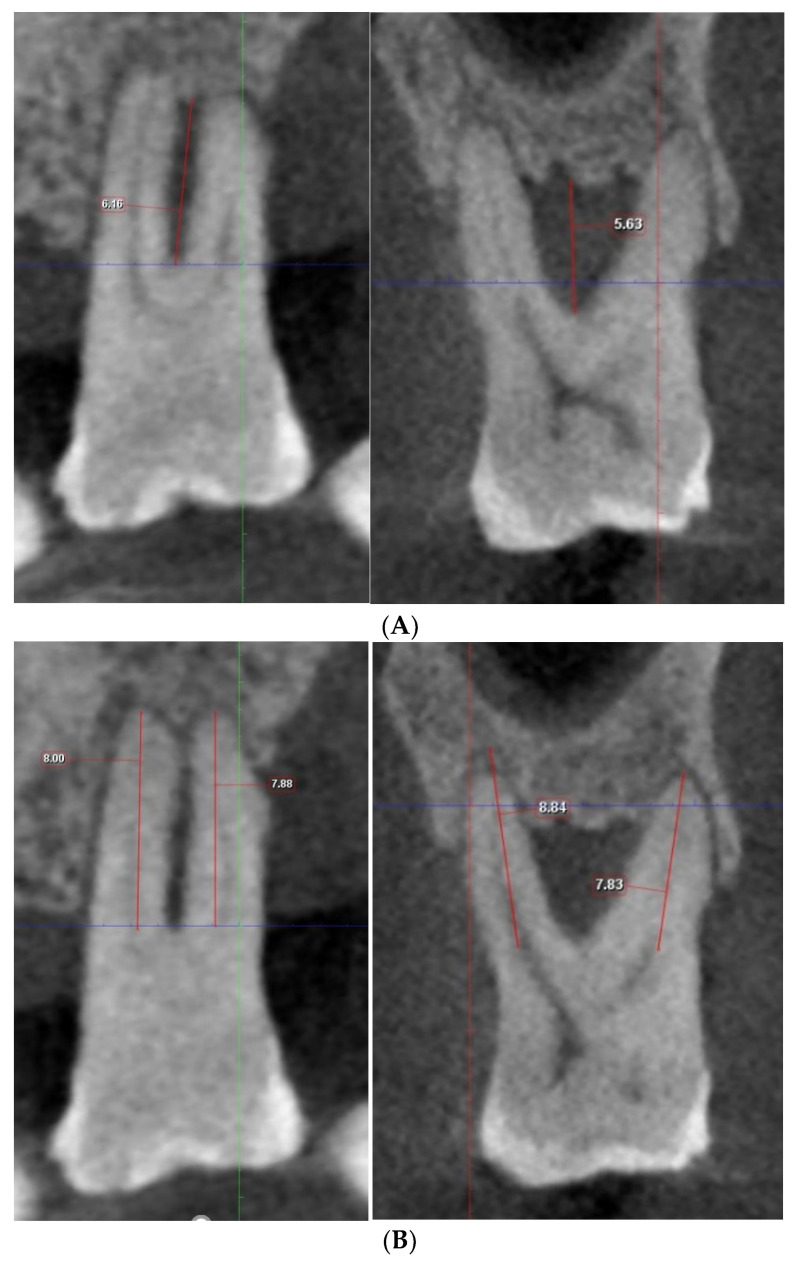
(**A**) Measurement of vertical depth of furcation and root length on Romexis (Planmeca). (**B**) By toggling the CBCT along the adjusted axis, the maximal vertical distance from the furcation fornix to the base of the defect and the length of the shorter root bounding the furcation were measured.

**Figure 3 diagnostics-13-00106-f003:**
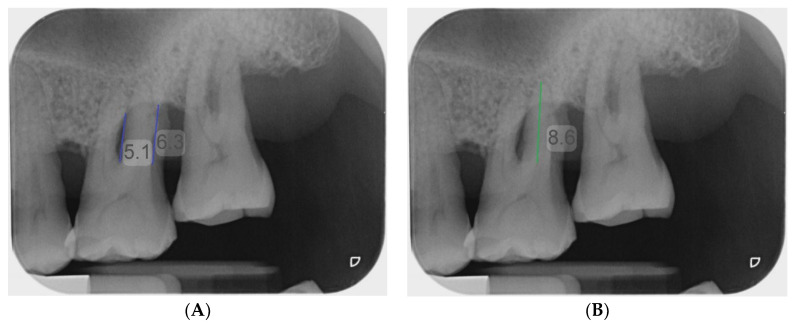
Measurement of vertical depth of furcation (**A**) and root length (**B**) on periapical radiograph using the measuring tool of Romexis (Planmeca).

**Table 1 diagnostics-13-00106-t001:** Intraclass correlation coefficient (ICC) for vertical depth of furcation involvement measured by conventional modality against CBCT.

Level 1 Examiner	1	2	3	4	5
All Teeth (95% CI)	0.52 (0.258, 0.712)	0.432 (0.142, 0.654)	0.402 (0.120, 0.628)	0.502 (0.234–0.700)	0.570 (0.322, 0.746)
Maxillary Molar (95% CI)	0.382 (−0.068, 0.700)	0.074 (−0.397, 0.499)	0.054 (−0.348, 0.461)	0.329 (−0.141, 0.671)	0.518 (0.131, 0.773)
Mandibular Molar (95% CI)	0.601 (0.213, 0.822)	0.775 (0.520, 0.904)	0.634 (0.290, 0.835)	0.702 (0.274,0.882)	0.559 (0.157, 0.800)
*“p*-value” (*F*-test)	0.174	0.002	0.022	0.042	0.424
**Level 2 Examiner**	**6**	**7**	**8**	**9**	**10**
All Teeth (95% CI)	0.133 (−0.147, 0.407)	0.331 (0.044, 0.573)	0.255 (−0.025, 0.509)	0.347 (0.039, 0.594)	0.287 (0.002, 0.537)
Upper Molar (95% CI)	−0.207 (−0.594, 0.252)	−0.001 (−0.348, 0.394)	0.382 (−0.019, 0.690)	0.204 (−0.224, 0.578)	0.022 (−0.361, 0.429)
Lower Molar (95% CI)	0.461 (−0.001, 0.754)	0.637 (0.271, 0.840)	0.079 (−0.230, 0.437)	0.523 (0.133, 0.776)	0.536 (0.003, 0.810)
“*p*-value” (*F*-test)	0.0435	0.016	0.196	0.137	0.0563
**Level 3 Examiner**	**11**	**12**	**13**	**14**	**15**
All Teeth (95% CI)	0.293 (−0.025–0.558)	0.442 (0.161, 0.658)	0.518 (0.247, 0.714)	0.352 (0.055, 0.594)	0.509 (0.243, 0.705)
Upper Molar (95% CI)	0.148 (−0.236, 0.522)	0.149 (−0.227, 0.521)	0.314 (−0.154, 0.662)	0.228 (−0.232, 0.603)	0.368 (−0.095, 0.694)
Lower Molar (95% CI)	0.387 (−0.108, 0.735)	0.567 (0.193, 0.800)	0.679 (0.349, 0.859)	0.468 (−0.103, 0.794)	0.619 (0.253, 0.830)
“*p*-value” (*F*-test)	0.240	0.0749	0.053	0.212	0.139

**Table 2 diagnostics-13-00106-t002:** Linear weighted kappa for vertical furcation subclassification determined by conventional modality against CBCT.

Level 1 Examiner	1	2	3	4	5
All Teeth (95% CI)	0.286 (0.049, 0.522)	0.308 (0.081, 0.534)	0.271 (0.018, 0.523)	0.214 (−0.026–0.454)	0.289 (0.051, 0.526)
Maxillary Molar (95% CI)	0.154 (−0.133, 0.441)	0.000 (−0.206, 0.206)	0.085 (−0.204, 0.373)	0.028 (−0.274, 0.330)	0.242 (−0.046, 0.531)
Mandibular Molar (95% CI)	0.394 (0.011, 0.799)	0.596 (0.242, 0.950)	0.394 (−0.011, 0.799)	0.394 (−0.011, 0.799)	0.559 (−0.135, 0.707)
“*p*-value” (*Z*-test)	0.342	0.004	0.222	0.154	0.866
**Level 2 Examiner**	**6**	**7**	**8**	**9**	**10**
All Teeth (95% CI)	0.071 (−0.154, 0.295)	0.315 (0.071, 0.559)	0.107 (−0.113, 0.398)	0.148 (−0.102, 0.594)	0.194 (−0.023, 0.411)
Upper Molar (95% CI)	−0.216 (−0.447, 0.015)	0.177 (−0.108, 0.462)	−0.053 (−0.270, 0.165)	0.058 (−0.251, 0.367)	−0.087 (−0.309, 0.136)
Lower Molar (95% CI)	0.369 (0.018, 0.721)	0.358 (−0.003, 0.719)	0.174 (−0.191, 0.538)	0.192 (−0.239, 0.623)	0.455 (0.114, 0.795)
“*p*-value” (*Z*-test)	0.006	0.444	0.295	0.621	0.009
**Level 3 Examiner**	**11**	**12**	**13**	**14**	**15**
All Teeth (95% CI)	0.113 (−0.096–0.323)	0.228 (0.017, 0.439)	0.350 (0.117, 0.582)	0.308 (0.091, 0.524)	0.287 (0.033, 0.541)
Upper Molar (95% CI)	−0.111 (−0.389, 0.167)	−0.087 (−0.182, 0.008)	0.179 (−0.109, 0.468)	0.110 (−0.153, 0.372)	0.143 (−0.188, 0.474)
Lower Molar (95% CI)	0.333 (−0.096, 0.571)	0.500 (0.122, 0.878)	0.490 (0.106, 0.874)	0.464 (0.141, 0.788)	0.406 (0.015, 0.797)
“*p*-value” (*Z*-test)	0.017	0.003	0.204	0.096	0.314

**Table 3 diagnostics-13-00106-t003:** Intraclass correlation coefficient (ICC) for assessment of reliability among the 15 examiners using the conventional modality in measuring vertical depth of furcation.

All Teeth (95% CI)	0.560 (0.446, 0.686)
Maxillary Molar (95% CI)	0.497 (0.343, 0.689)
Mandibular Molar (95% CI)	0.573 (0.415, 0.750)
Isolated Class II at B(Buccal) (95% CI)	0.491 (0.199, 0.934)
Isolated Class II at M/D(mesio/distal) (95% CI)	0.470 (0.250, 0.796)
At Least 2 Class II/Class III (95% CI)	0.624 (0.395, 0.877)
Isolated Class II at B/L(bucco/lingual) (95% CI)	0.544 (0.334, 0.807)
Class II at B/L or Class III (95% CI)	0.505 (0.297, 0.783)

(1) The *p*-value of the *F*-test analysis between the maxillary molars and mandibular molars was 0.362. (2) In maxillary molars, the *p*-value of the *F*-test analysis between the isolated buccal class II furcation group and isolated mesial/distal class II furcation group was 0.537. (3) In maxillary molars, the *p*-value of the *F*-test analysis between the isolated mesial/distal class II furcation group and 2 class II furcation/class III furcation group was 0.331. (4) In maxillary molars, the *p*-value of the *F*-test analysis between the isolated buccal class II furcation group and 2 class II furcation/class III furcation group was 0.333. (5) In mandibular molars, the *p*-value of the *F*-test analysis between the isolated buccal/lingual class II furcation group and 2 class II furcation/class III furcation group was 0.452.

**Table 4 diagnostics-13-00106-t004:** The relationship between bias in vertical depth of furcation involvement measurement with tooth position, furcation involvement distribution, and examiner experience in the linear mixed model (unadjusted).

Parameter: Tooth Position	Estimate	95% Confidence Interval		*p*-Value	
		Lower Bound	Upper Bound		Pairwise Comparison
Intercept	−0.408	−0.895	0.079	0.098	
Maxillary molar	0.169	−0.520	0.857	0.623	
Mandibular molar	0				
**Parameter: Furcation Involvement Distribution**					
Intercept	−0.238	−0.854	0.379	0.439	[1] > [2]
				0.022	
Isolated maxillary buccal class II furcation involvement	1.216	0.064	2.369	0.039	
Isolated maxillary mesial/distal class II furcation involvement	−0.792	−1.716	0.132	0.091	
At least two maxillary class II furcation involvements or class III furcation involvement	0.172	−0.752	1.097	0.707	
Isolated mandibular buccal/lingual class II furcation involvement	−0.341	−1.212	0.531	0.433	
Buccal and lingual class II furcation involvement or class III furcation involvement	0				
**Parameter: Examiner Experience Level**		**Lower Bound**	**Upper Bound**		
Intercept	−0.405	−0.756	−0.054	0.025	[Level 1] > [Level 2] = [Level 3]
				<0.01	
Level 1 Examiner Experience (recent graduates with less than one year of postgraduate experience)	0.219	0.086	0.352	0.001	
Level 2 Examiner Experience (postgraduate clinicians with at least two years of postgraduate experience)	−0.165	−0.334	0.004	0.055	
Level 3 Examiner Experience (specialist periodontists)	0				

**Table 5 diagnostics-13-00106-t005:** Descriptive statistics and one-way ANOVA comparing the vertical depth of furcation involvement in different furcation involvement distribution subgroups as measured on CBCT.

	N	Mean (mm)	Std. Deviation	Minimum	Maximum	*p*-Value
Furcation Distribution						<0.001
Isolated maxillary buccal class II furcation involvement	4	4.5475	1.26800	3.06	6.10	
Isolated maxillary mesial/distal class II furcation involvement	8	2.5675	0.82794	1.58	3.81	
At least two maxillary class II furcation involvements or class III furcation involvement	8	3.6463	1.42269	1.61	6.19	
Isolated mandibular buccal/lingual class II furcation involvement	10	1.9250	0.57161	1.24	3.00	
Mandibular buccal and lingual class II furcation involvement or class III furcation involvement	10	3.1480	0.61387	2.08	4.13	
Total	40	2.9658	1.20115	1.24	6.19	

**Table 6 diagnostics-13-00106-t006:** The relationship between bias in vertical depth of furcation involvement measurement, furcation involvement distribution, and examiner experience level in the linear mixed model (adjusted for gold standard vertical depth of furcation involvement).

Parameter	Estimate	95% Confidence Interval			Bonferroni Multiple Comparisons
		Lower Bound	Upper Bound	*p*-Value	
Intercept	−2.117	−3.200	−1.034	<.001	
Furcation Involvement Distribution				0.228	
Isolated maxillary buccal class II furcation involvement	0.399	−0.664	1.461	0.451	
Isolated maxillary mesial/distal class II furcation involvement	−0.486	−1.287	0.315	0.226	
At least two maxillary class II furcation involvements or class III furcation involvement	−0.116	−0.913	0.680	0.768	
Isolated mandibular buccal/lingual class II furcation involvement	0.358	−0.466	1.182	0.384	
Mandibular buccal and lingual class II furcation involvement or class III furcation involvement	0	.	.	.	
Examiner Experience Level				<0.001	[Level 1] > [Level 2] = [Level 3]
Level 1 Examiner Experience (recent graduates with less than one year of postgraduate experience)	0.225	0.092	0.358	0.001	
Level 2 Examiner Experience (postgraduate clinicians with at least two years of postgraduate experience)	−0.162	−0.331	0.007	0.061	
Level 3 Examiner Experience (specialist periodontists)	0	.	.	.	
Gold Standard Vertical Depth	0.573	0.273	0.874	<0.001	

## Data Availability

Not applicable.

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
