# Peer review of "Comparison between Conventional Modality Versus Cone-Beam Computer Tomography on the Assessment of Vertical Furcation in Molars"

_diagnostics, 2022, doi:10.3390/diagnostics13010106_

Round 1

Reviewer 1 Report

Congratulations for this nice retrospective study. The aim is pertinent and atual

The introduction could refer to a recent Systematic review about this subject: Jolivet G, Huck O, Petit C. Evaluation of furcation involvement with diagnostic imaging methods: a systematic review. Dentomaxillofac Radiol. 2022 Dec 1;51(8):20210529. doi: 10.1259/dmfr.20210529.

Materials and Methods is well designed and very well described.

Results are well described and exhaustive.

Discussion must be improved, particularly in comparing the present results to other similar studies, as there are some published: ex Zhang W, Foss K, Wang BY. A retrospective study on molar furcation assessment via clinical detection, intraoral radiography and cone beam computed tomography. BMC Oral Health. 2018 May 3;18(1):75. doi: 10.1186/s12903-018-0544-0.

The third paragraph (lines 299 to 305) is Results and should be eliminated or altered to present another kind of information.  

Author Response

We express our most sincere gratitude for your instructive comments and suggestions on our manuscript Comparison between Conventional Modality versus Cone-Beam Computer Tomography on the Assessment of Vertical Furcation in Molars.

We have studied the reviewers’ comments carefully and tried our best to revise our manuscript according to the comments. We look forward to hearing from you soon.

Thank you and best regards.

Yours sincerely,

Corresponding author: Georgios Pelekos

Responses to the reviewers’1 comments:

Referee: 1
Comment 1- The introduction could refer to a recent Systematic review about this subject: Jolivet G, Huck O, Petit C. Evaluation of furcation involvement with diagnostic imaging methods: a systematic review. Dentomaxillofac Radiol. 2022 Dec 1;51(8):20210529. doi: 10.1259/dmfr.20210529.

Response: Thank you for your valuable comment.

In the revised manuscript, we added a few sentences and this excellent reference in the introduction part (77-80)

Comment 2- Discussion must be improved, particularly in comparing the present results to other similar studies, as there are some published: ex Zhang W, Foss K, Wang BY. A retrospective study on molar furcation assessment via clinical detection, intraoral radiography and cone beam computed tomography. BMC Oral Health. 2018 May 3;18(1):75. doi: 10.1186/s12903-018-0544-0

Response: Thank you for pointing this out. We added the following two references;

  1. Zhang W, Foss K, Wang BY. A retrospective study on molar furcation assessment via clinical detection, intraoral radiography and cone beam computed tomography. BMC Oral Health. 2018 May 3;18(1):75.
  2. Assiri H, Dawasaz AA, Alahmari A, Asiri Z. Cone beam computed tomography (CBCT) in periodontal diseases: a Systematic review based on the efficacy model. BMC Oral Health. 2020 Jul 8;20(1):191

In the discussion part, we added a new paragraph (291-295) and a few more new sentences (311-313) to improve this part.

Comment 3- The third paragraph (lines 299 to 305) is Results and should be eliminated or altered to present another kind of information

Response: Thank you very much for pointing out our inappropriate logical flow in the previous manuscript. We removed this paragraph.

Reviewer 2 Report

In the radiographic section, authors state that exposure times varied from 0.01 to 2s – for PSP plate technology, these values are in a high range (200x times difference in time settings for intraoral, which is unusual and requires some comment). The statement “images we  developed” is inappropriate as PSP plates are laser scanned to reveal a latent image.  

CBCT settings for Planmeca and Morita CBCT units should be provided – particularly the voxel size obtained.

In the examiners and image analysis, authors stated: “The examiners were briefed on the study procedures and the furcation classification 148 systems used in the assessment” does it mean they were taught about how to handle the intraoral x-ray image and cbct volume? Were the examiners calibrated prior to performing their own measurements? This might be a key factor when looking into the results section where there is a statement: “Examiner experience level also appears to affect the bias in measuring the vertical 256 depth of furcation involvement (p<0.01)”. Is the experience term refers to the experience in periodontal treatment/knowledge or the experience in handling/measuring data on CBCT/ intraoral images? As the presented study is based on the given charts. I suppose that the clinical examination was performed before the study, perhaps by other doctors. So this implies that authors, when stating “experience,” should possibly explain how the level 1, level 2, and level 3 examiners are gaining their confidence in radiology rather than in periodontology.

In the conclusion section, perhaps the authors can provide their recommendation about when CBCT should be involved as a part diagnostic procedure. Should we all do it every time we can diagnose furcation involvement? Perhaps in maxilla? Or when a clinical diagnosis suggests class III furcation involvement? 

Author Response

We express our most sincere gratitude for your instructive comments and suggestions on our manuscript Comparison between Conventional Modality versus Cone-Beam Computer Tomography on the Assessment of Vertical Furcation in Molars.

We have studied the reviewers’ comments carefully and tried our best to revise our manuscript according to the comments. We look forward to hearing from you soon.

Thank you and best regards.

Yours sincerely,

Corresponding author: Georgios Pelekos

Responses to the reviewers’2 comments:

Comment 1: In the radiographic section, authors state that exposure times varied from 0.01 to 2s – for PSP plate technology, these values are in a high range (200x times difference in time settings for intraoral, which is unusual and requires some comment). The statement “images we developed” is inappropriate as PSP plates are laser scanned to reveal a latent image.

Response: Thank you for your valuable comment. We corrected the typo concerning the exposure time (112) and removed as correctly pointed the word developed.

Comment 2- CBCT settings for Planmeca and Morita CBCT units should be provided – particularly the voxel size obtained

 Response: Thank you for pointing this out. We added the missing settings information in lines 115-116.

Comment 3- In the examiners and image analysis, authors stated: “The examiners were briefed on the study procedures and the furcation classification 148 systems used in the assessment” does it mean they were taught about how to handle the intraoral x-ray image and cbct volume? Were the examiners calibrated prior to performing their own measurements? This might be a key factor when looking into the results section where there is a statement: “Examiner experience level also appears to affect the bias in measuring the vertical 256 depth of furcation involvement (p<0.01)”. Is the experience term refers to the experience in periodontal treatment/knowledge or the experience in handling/measuring data on CBCT/ intraoral images? As the presented study is based on the given charts. I suppose that the clinical examination was performed before the study, perhaps by other doctors. So this implies that authors, when stating “experience,” should possibly explain how the level 1, level 2, and level 3 examiners are gaining their confidence in radiology rather than in periodontology

Response: Thank you very much for pointing out a critical area since teaching clinical exposure and experience may vary considerably among different faculties and different departments, thus reflecting on the issue and definition of ‘operator experience ‘.We added a new paragraph elaborating on that matter(151-154)

Comment 4-In the conclusion section, perhaps the authors can provide their recommendation about when CBCT should be involved as a part diagnostic procedure. Should we all do it every time we can diagnose furcation involvement? Perhaps in maxilla? Or when a clinical diagnosis suggests class III furcation involvement? 

Response: According to the reviewer’s comment, we added a few sentences to reflect on clinical recommendations (291-295) in the conclusion part.